# Perceptions of Australia's e-cigarette regulations and recommendations for future reforms: a qualitative study of adolescents and adults

Mary-Ellen E Brierley [ID],[1] Sean J L Yaw,[1,2] Michelle I Jongenelis [ID] [1]

[1]Melbourne Centre for Behaviour Change, Melbourne School of Psychological Sciences, The University of Melbourne, Melbourne, Victoria, Australia
[2]Melbourne School of Population and Global Health, The University of Melbourne, Melbourne, Victoria, Australia

**Correspondence to**
Dr Michelle I Jongenelis;
michelle.jongenelis@unimelb.edu.au

## ABSTRACT

**Objective** To assess public perceptions of the effectiveness of e-cigarette regulations in minimising use among adolescents and those who have never smoked. Specifically, we explored (1) perceived effectiveness of current regulations relating to e-cigarettes and (2) ideas for further regulations that could reduce use.

**Design and participants** Focus groups (n=16) were conducted with Australian adolescents (14–17 years), young adults (18–24 years) and adults (25–39 years). Groups were stratified by age, gender and e-cigarette use status. Data were analysed using reflexive thematic analysis.

**Setting** Focus groups were conducted in-person in two major Australian cities.

**Results** Groups lacked a comprehensive understanding of e-cigarette regulations in Australia. When informed of these regulations, half of the groups considered the prescription model for nicotine e-cigarette products to be effective when enforced appropriately. Almost all groups considered access to non-nicotine products problematic. All groups suggested a range of demand reduction regulations, including plain packaging, health warnings, flavour restrictions and increased vape-free areas. Most groups (predominantly those who had never vaped) also recommended supply reduction regulations such as banning all e-cigarettes. The need for supply reduction measures to include addiction and mental health supports was discussed.

**Conclusions** The regulations recommended by participants largely align with those that are to be introduced in Australia, indicating that these reforms are likely to be accepted by the public. Ensuring these reforms are complemented by formal supports for young people experiencing nicotine dependence and related mental health concerns is critical.

## STRENGTHS AND LIMITATIONS OF THIS STUDY

⇒ This was the first study to explore perceptions of e-cigarette regulation effectiveness, with prior work in this space assessing regulation support only.
⇒ We recruited participants across the community and stratified groups by age, gender and vaping status to gather a broad range of perspectives and explore differences between groups.
⇒ Due to the emergent nature of the coding process, only one researcher coded the data, which prevented the calculation of intercoder reliability.
⇒ Findings only represent the perspectives of the 139 participants who attended the focus groups and caution should be exercised when generalising to the broader population.
⇒ As this research was conducted in one country, future research could explore perceptions of the effectiveness of e-cigarette regulations in other jurisdictions.

## INTRODUCTION

Recent years have seen global increases in the prevalence of e-cigarette use, particularly among youth.[1 2] In Australia, the context of the present study, e-cigarette use has increased significantly since reporting measures were introduced in 2013,[3] with the number of adolescents and young adults who report having used an e-cigarette in the past month increasing approximately fivefold since 2018.[4] These increases are concerning given vaping has been found to be associated with several physical and mental health harms and subsequent initiation of tobacco cigarette smoking.[5–7] The increasing prevalence of e-cigarette use has largely occurred in the context of decreasing tobacco cigarette use, although recent data have observed an increase in tobacco smoking among adolescents and young adults for the first time in three decades.[4]

The substantial increase in e-cigarette use has prompted calls for tightened regulation of the devices to reduce uptake and minimise potential health risks, especially among youth and those who have never smoked.[8 9] Since 1 October 2021, nicotine-containing e-cigarettes and related products have been legally available to adults only via prescription from a medical doctor for the purposes of smoking cessation.[10] E-cigarettes that do not contain nicotine are less restricted and

may be sold by retailers to those aged 18+ years in all states and territories except Western Australia.[10] The supply of e-cigarettes—regardless of nicotine content—to individuals under 18 years of age has never been permitted.

Despite these restrictions, increases in use continue to be observed. This is likely due to (1) a lack of appropriate controls at the Australian border and (2) the importation and sale of non-nicotine e-cigarette products remaining legal, which has led to mass importation and high availability of non-nicotine and incorrectly labelled nicotine e-cigarette products on the Australian market.[11] To address these issues, Australia's Federal Government announced plans in May 2023 to introduce regulations that prohibit the importation of both nicotine and non-nicotine e-cigarettes for non-therapeutic use.[12] Restrictions on flavourings and the introduction of pharmaceutical-like packaging were also proposed to reduce the appeal of e-cigarette products. These regulations will be implemented on 1 March 2024.[13]

Public acceptability of regulations is an important consideration when developing and implementing policy, with research typically operationalising acceptability as the extent to which a regulation is supported.[14 15] Research conducted in Australia has found high levels of support for the introduction of tighter regulations on e-cigarettes, with the vast majority of adults endorsing (1) restricted access to and advertising of e-cigarettes and (2) the expansion of vape-free public areas.[16] Consistent with the notion that support for a particular regulation is moderated by the extent to which that regulation will restrict one's behaviour,[14] support for e-cigarette regulations has been found to differ based on vaping status. E-cigarette users typically (1) support non-restrictive regulations, such as making e-cigarette products available to all adults via retail stores and allowing vaping in smoke-free areas and (2) oppose regulations that restrict the supply of the devices.[9 17] By contrast, those who have never vaped tend to oppose measures that would result in e-cigarettes being readily available.[9]

Given support for e-cigarette regulations appears to be heavily moderated by vaping status, considering alternative measures of acceptability is critical to obtaining a more objective account of community views. Perceived effectiveness is one such alternative and an important component of acceptability.[18] Greater perceived effectiveness of a government-initiated health regulation is predictive of more favourable community attitudes toward that regulation and greater compliance,[14 19] suggesting that perceived effectiveness of e-cigarette regulation may provide useful information on the likely outcome of regulation implementation. Despite this, there is a lack of research exploring Australians' perceptions of the effectiveness of current regulations relating to e-cigarettes and their ideas for further regulations that are likely to be effective at reducing e-cigarette use. Accordingly, this study aimed to explore adolescents', young adults' and adults' perspectives on:

1. The effectiveness of current e-cigarette regulations in Australia.
2. Regulations they believe could minimise e-cigarette use, particularly among adolescents and those who have never smoked.

## METHOD

### Recruitment and sample

A social research agency was commissioned to recruit a sample of Australians aged 14–39 years to participate in 1 of 16 focus groups (FGs) conducted in Melbourne and Sydney. Groups were stratified by (1) age (14–15 years, 16–17 years, 18–24 years and 25–39 years), (2) gender (women and men) and (3) e-cigarette user status (current/past vapers and those who had never vaped). Table 1 presents the composition of each group. Groups ranged in size from 6 to 10 participants.

### Procedure

FGs were conducted in March 2023 (prior to the Government's announcement) and were approximately 70 min in duration (range: 57–88 min). All groups were facilitated by MIJ, a principal research fellow with a PhD in clinical psychology. Participants completed a short survey while waiting for their FG to begin. Items in the survey assessed participants' sociodemographic characteristics (eg, gender, age).

A semistructured interview guide comprising open-ended questions was followed. To orient participants to the topics being discussed, all groups began with initial questions exploring participants' experiences with e-cigarettes and knowledge of current regulations. These were then followed by questions exploring (1) participants' opinions of current regulations and (2) their ideas for what could be done in Australia to effectively reduce e-cigarette use, especially among adolescents and those who have never smoked.

For participants aged <16 years, consent was also obtained from a caregiver. Participants were reimbursed $A120 for their time and the costs associated with FG attendance. Caregivers were reimbursed $A30. To ensure adolescents felt comfortable speaking openly about vaping, caregivers were not present during the FGs.

### Patient and public involvement

No involvement.

### Data analysis

FGs were audiorecorded by the research team and transcribed verbatim by an independent and International Organization for Standardization-accredited transcription agency. Transcripts were then imported into NVivo for coding and analysis. As this research was data driven rather than theory driven, an inductive (ie, emergent) approach to thematic analysis was adopted.[20] One researcher (MEB) analysed all transcripts according to the iterative steps of the reflexive thematic analysis framework detailed by Braun and Clarke.[21] This

**Table 1** Focus group characteristics

| Group | N | Age | Gender | E-cigarette user status | Location |
|---|---|---|---|---|---|
| 1 | 9 | 14–15 years | Women | Current or past | Sydney |
| 2 | 10 | | | Never | Melbourne |
| 3 | 10 | | Men | Current or past | Sydney |
| 4 | 10 | | | Never | Melbourne |
| 5 | 9 | 16–17 years | Women | Current or past | Melbourne |
| 6 | 9 | | | Never | Sydney |
| 7 | 7 | | Men | Current or past | Melbourne |
| 8 | 10 | | | Never | Sydney |
| 9 | 7 | 18–24 years | Women | Current or past | Melbourne |
| 10 | 8 | | | Never | Sydney |
| 11 | 6 | | Men | Current or past | Melbourne |
| 12 | 10 | | | Never | Sydney |
| 13 | 9 | 25–39 years | Women | Current or past | Melbourne |
| 14 | 9 | | | Never | Melbourne |
| 15 | 9 | | Men | Current or past | Melbourne |
| 16 | 7 | | | Never | Melbourne |

included data familiarisation, code generation, theme generation, theme review, defining and naming themes, and combining the analysis of themes in the context of a report.[21] Throughout this process, a series of 'critical friends' meetings was held.[22] During these meetings, MIJ and MEB reviewed the coding hierarchy, refined the identified codes and generated key themes based on these codes.

Initial codes were generated that organised the data into the following broad topics: (1) beliefs about current regulations and (2) recommendations for potential regulations. For the former, child codes were created that further organised the data into the topics of (1) beliefs about the prescription model and (2) beliefs about the availability of non-nicotine e-cigarette products. Within each of these child codes, further codes were generated that organised the data based on (1) positive sentiment and (2) negative sentiment.

In terms of regulation recommendations, child codes were created for each of the ideas generated by participants (eg, price increases, plain packaging, product warning labels). Codes featuring similar content were then merged (eg, 'plain packaging' and 'product warning labels' were merged to form a 'product packaging' code). These codes were then refined into the categories of (1) supply factors and (2) demand factors. Descriptive labels were assigned to all codes. Please see online supplemental material for the final coding hierarchy.

The Matrix Coding Query function in NVivo was used to explore any similarities and differences in data according to (1) age group (adolescents (14–17 years) cf. young adults (18–24 years) cf. adults (25+ years)), (2) gender (women cf. men) and (3) vaping status (current/past vapers cf. those who had never vaped). Quotes are

provided throughout the results section to highlight specific findings of interest. Each participant quote is followed by details of the FG of which the participant was a part: FG number (eg, FG #1); adolescents or young adults or adults aged 25+ years; W=women or M=men; V=vapers or NV=those who had never vaped.

## RESULTS

All groups had some experience with e-cigarettes. Among those who had vaped in the past or were current vapers, direct experiences with e-cigarettes were reported. Among those who had never vaped, experiences were indirect and typically involved observing e-cigarette use at school or in the workplace, when out socialising, and by friends or family members. All groups commented that e-cigarette use was increasing and the devices were 'everywhere'.

Understanding of Australia's current regulations relating to e-cigarettes was low overall, with less than half of all groups mentioning the prescription model for nicotine e-cigarette products. Instead, many groups noted that e-cigarettes were illegal, with some groups specifying that e-cigarettes containing nicotine were illegal. There was a great deal of uncertainty observed, with participants in many groups noting that they were 'guessing' when reporting on the regulations. All groups commented on the high availability of e-cigarettes in the community, with most groups reporting that this gave the impression there were no regulations:

I feel that there are no regulations in Australia…I feel there is none because of the shops. I also see that there are neon boards with displays at night. You can

clearly see that the vape is available. So, everyone has the access to it…it's just like the government is giving them permission to display it.—FG#14, adults aged 25+years, W, NV.

## Perspectives on current regulations

After being informed of the current regulations by the facilitator, groups discussed their perspectives on the prescription model for nicotine e-cigarette products and the availability of non-nicotine e-cigarette products. Half of the groups considered the prescription model to be an effective means of reducing access to e-cigarettes, and thus reducing e-cigarette use. Concerns about the model were raised, however (outlined in detail below). In terms of non-nicotine e-cigarette availability, most groups were unaware that such products existed. When informed that these products were available for sale to those aged 18+ years, most groups noted that this availability may increase overall e-cigarette use. Some groups also reported lacking an understanding of why a market would exist for non-nicotine e-cigarettes, noting that 'I don't feel a lot of people would buy non-nicotine ones because they want the nicotine.'—FG#5, adolescents, W, V.

## Perspectives on the availability of nicotine e-cigarette products (ie, the prescription model)

Half of all groups (predominantly those who had never vaped) considered the prescription model to be effective at reducing e-cigarette use among youth and adults who have never smoked. Participants noted that the model made it difficult for these cohorts to access e-cigarettes:

It will stop people being able to buy them, thinking that's okay, and then having access.—FG#9, young adults, W, V.

Potential issues with the prescription model were raised. In some groups, participants did not believe e-cigarettes were effective smoking cessation aids and reported that the devices have the potential to make people more addicted to nicotine. Accordingly, a medical model was not considered appropriate. Few groups (all of which comprised vapers) discussed potential barriers to accessing a prescription. These barriers included medical practitioner hesitancy to prescribe nicotine e-cigarette products and the financial costs associated with a medical consultation. The prescription model being difficult to enforce and thus easy to circumvent was also raised as a potential issue.

I don't think it's going to help you quit. It's just going to make you more addicted to it.—FG#1, adolescents, F, V.

There's such little knowledge about what's actually in these. Why, as a GP, would you recommend that?—FG#15, adults aged 25+years, M, V.

Who's asking you for a prescription? Is it people in the street who are like, 'Hey, you're smoking that

vape, do you have your prescription on you?'—FG#13, adults aged 25+years, W, V.

## Perspectives on the availability of non-nicotine e-cigarette products

Opinions on non-nicotine e-cigarette products being available to all adults in Australia were largely negative. Groups voiced concerns that the availability of these products undermines attempts to reduce vaping as these products may be perceived as safe. It was also noted that non-nicotine e-cigarette products lead to addiction and are a gateway to nicotine e-cigarette use. Finally, groups voiced mistrust in the labelling of e-cigarette products given the differing regulations between non-nicotine and nicotine varieties.

It's also kind of dumb that they only ban the nicotine one because it makes you think that the other one's safe, when it's probably just as bad.—FG#3, adolescents, M, V.

What's stopping an importer from just putting a sticker that says no nicotine?—FG#16, adults aged 25+years, M, NV.

Few positive perspectives on the availability of non-nicotine e-cigarettes were observed. Those voicing a positive perspective (1) noted that non-nicotine e-cigarettes may be used to quit smoking and/or (2) endorsed libertarianism.

## Recommendations for potential regulations

When discussing regulatory action that could be taken to effectively reduce e-cigarette use in Australia, groups cited both 'demand reduction' and 'supply reduction' reforms. Demand reduction reforms were most frequently mentioned, with all groups citing such measures as being important. Perspectives on supply reduction reforms were mixed: groups comprising adolescents, young adults and those who had never vaped recommended regulations that would heavily restrict e-cigarette availability in the community whereas vapers tended to recommend policies that would increase availability for adults. Suggestions for future regulations were sometimes accompanied by discussions on the addictive nature of e-cigarettes, the link between use and mental health issues, and the importance of ensuring regulations were introduced alongside community-funded or government-funded support programmes. The need for addiction and mental health supports to reduce e-cigarette use were typically raised by adolescents (both vapers and those who had never vaped).

## Demand reduction measures

Specific demand reduction measures cited by FG participants included introducing plain packaging, including health warnings on e-cigarette products, restricting flavours, increasing the cost of e-cigarette products, increasing the number of vape-free public areas and restricting advertising. All groups voiced the belief that

the introduction of plain packaging and/or health warning messages to mirror existing regulations for tobacco cigarettes would reduce use. Almost all groups reported that prohibiting flavourings and increasing the cost of e-cigarettes would reduce use, particularly among young people. Flavourings were considered highly appealing, and the devices were considered inexpensive, and therefore, affordable to youth.

> They do make vapes look appealing. They're all fluorescent colours, they look good whereas cigarettes are in plain packaging now. I would create consistency across products that, to me, appear more or less the same.—FG#15, adults aged 25+years, M, V.

> If it didn't taste like anything or wasn't nice, if it was yuck, then I feel like it would solve half the problem.—FG#5, adolescents, W, V.

> They're [young people] buying a vape now because it's $15 and it's accessible and $15 is nothing. But if you upped that price to $60, people will question if it's worth it.—FG#10, young adults, W, NV.

Most groups discussed increasing the number of vape-free areas and improving signage around vape-free zones. Most groups also voiced concerns that e-cigarettes are marketed towards young people and highly visible in the community due to retail shop advertising. As such, regulations that restrict visibility and advertising were recommended.

> Smoke-free zones, or more of them, because they're pretty strong on cigarettes, like 'no smoking in this area', but people are like 'we're going to vape', and it's the norm, but it really shouldn't [be]—FG#11, young adults, M, V.

> The outdoor marketing should be banned. There shouldn't be night lights which are clearly saying which shops have access to it… just like all the other cigarette brands that are being sold.—FG#14, adults aged 25+years, W, NV.

### Supply reduction measures

Perspectives on supply reduction reforms were mixed and appeared to be influenced by factors other than perceived effectiveness, such as libertarian views and participant age and vaping status. Some groups (mostly adolescents and those who had never vaped) believed banning e-cigarettes would be most effective and suggested (1) banning the importation of all e-cigarettes, (2) banning all disposable e-cigarettes and (3) making e-cigarettes less available to children and adolescents. Few groups (mostly vapers) recommended legalising the products in a manner consistent with tobacco cigarettes. Across all groups, the importance of enforcement and taking action against those selling the products illegally was evident:

> I think just actually enforce the law…If you can just go to any tobacco shop and get it, what even is the

point in the law if it's not being enforced…—FG#8, adolescents, M, NV.

Some differences were observed by age group and vaping status. All adolescent groups discussed supply reduction measures, while half of the young adult and adult groups did so. In terms of vaping status, supply factors were raised by both vapers and those who had never vaped, but those who had never vaped endorsed such measures more strongly. In addition, those who had never vaped typically focused on the importance of reducing use across all age groups ('I personally think they just shouldn't be allowed'.—FG#12, young adults, M, NV), whereas vapers focused on reducing use among adolescents ('They could show IDs and make sure they are 18.'—FG#13, adults aged 25+years, W, V). Adult and young adult vapers typically discussed a model that mirrors what is currently in place for tobacco cigarettes, with adult access to e-cigarettes legalised. A focus of these discussions was libertarianism, with some participants noting that adults have a right to engage in vaping:

> Make it legal. I can still go to McDonald's every day and feed my kid McDonald's and it's fine. I'm not going to say that I want to give my kid e-cigs or anything, but as an adult let me adult as long as I'm not hurting someone else.—FG#15, adults aged 25+years, M, V.

Finally, some vapers believed that legalisation would reduce e-cigarette use as it would curtail the black market. Some also believed there was hypocrisy in e-cigarettes being less available than cigarettes and noted that there needed to be more evidence on the health impacts of vaping to justify supply restrictions.

## DISCUSSION

E-cigarette use is increasing globally, particularly among adolescents and young people.[1 2] This has prompted calls for all levels of government to take regulatory action to minimise uptake of the devices to protect tobacco control efforts.[9 11] Given acceptability is an important consideration when implementing new regulations,[15] this study examined adolescents', young adults' and adults' (1) perceptions of the effectiveness of Australia's e-cigarette regulations and (2) recommendations for regulations that have the potential to reduce e-cigarette use, particularly among adolescents and those who have never smoked. Results offer key insights into public perceptions of Australia's regulatory framework for e-cigarette products. They also offer insights for jurisdictions currently considering the implementation of a prescription model (eg, Scotland) and disposable e-cigarette bans (eg, the UK and European Union member states such as Ireland, Germany and France).

FG participants lacked a comprehensive understanding of e-cigarette regulations in Australia, with few groups aware of the prescription model for nicotine e-cigarette products. Participants reported that the widespread

availability of e-cigarettes and the advertising practices of retail stores that sold e-cigarette products communicated a lack of regulation. Accordingly, the need to enforce existing regulations to improve their effectiveness and reduce the availability of nicotine and non-nicotine e-cigarettes was widely discussed. This was particularly evident when discussing the prescription model for nicotine products, with half of the groups considering the model effective at restricting access to e-cigarettes but only when regulations relating to the model were enforced.

Several other issues relating to the prescription model were raised, although by only some groups. These included potential barriers to accessing a prescription, such as cost and practitioner hesitancy to prescribe. Hesitancy among medical practitioners to recommend and/or prescribe e-cigarettes has been identified in previous research.[23 24] Consumers of e-cigarettes have also raised concerns about practitioner hesitancy and the impact of this on access.[25] Given the effectiveness of the prescription model is dependent on medical practitioners being open to prescribing e-cigarettes to smokers who wish to quit but have been unable to do so with first-line treatments, it is critical that practitioners are supported in their practice to prescribe nicotine e-cigarette products when use is clinically indicated.

FG participants were generally unaware of the existence of non-nicotine e-cigarettes and raised several concerns regarding these products when informed of their presence on the Australian market. The most frequently cited concern was that ready availability of these products communicates a message that they are safe. Other concerns included (1) nicotine products being labelled as non-nicotine products to bypass regulations and (2) non-nicotine product use acting as a gateway to nicotine use. These concerns are supported by the literature. Previous research has found that 60% of e-liquids affixed with a label that claims they are nicotine-free actually contain nicotine.[26] There is also evidence that use of non-nicotine products leads to use of nicotine products, with a recent study finding that approximately 25% of those who exclusively use non-nicotine products will transition to using nicotine products after 1 year.[27] The concerns raised regarding non-nicotine e-cigarettes, and the finding that many in the sample questioned why these products existed and their utility, suggests that plans to restrict access to such products will likely be accepted by the general public.

Concerning potential future regulations, FG participants recommended several demand and supply reduction measures that they believed would be effective at reducing e-cigarette use. Demand reduction measures were most frequently cited and included placing health warnings on e-cigarette products, restricting flavours, increasing the cost of e-cigarette products, increasing the number of vape-free public areas and restricting advertising. Given many of these reforms were announced by the Australian Government in May 2023,[12] it can be reasonably expected that implementation of the proposed reforms will be met with little objection from the general public.

Recommended regulations for reducing the supply of e-cigarettes in the community included banning all e-cigarette products, banning the importation of nicotine e-cigarette products, banning all disposable e-cigarettes, reducing access by adolescents and maintaining adoption of the prescription model. Supply reduction regulations were typically suggested by adolescents and those who had never vaped, whereas adults who vaped tended to adopt a liberal approach and recommended models of availability like those in place for tobacco cigarettes. The differing views of adults who vape compared with other participant groups may be explained by perceived threats to existing behaviours or freedoms and is consistent with previous research showing that e-cigarette users typically oppose regulations that restrict the supply of the devices.[9 17] Communications highlighting the historical mistake of tobacco cigarettes being made consumer products could be used to inform vapers of the risks associated with regulating e-cigarettes in a manner that is consistent with tobacco products.[25]

It is encouraging that adolescents and young adults, regardless of vaping status, were largely supportive of a range of demand and supply reduction regulations to address e-cigarette use. These findings suggest Australian youth wish to quit vaping and consider the range of measures proposed, including reduced availability, as critical. This is consistent with other studies that have found high intentions to quit e-cigarettes among young people.[28 29] However, consideration should be given during implementation of supply reduction regulations to ensure appropriate supports are offered to those who are addicted to e-cigarettes or who are using e-cigarettes to cope with mental health struggles, with adolescents in the sample expressing concerns about nicotine dependence and poor mental health.

### Limitations and strengths

This study had some limitations which should be considered when interpreting the findings. First, due to the emergent nature of the coding process, only one researcher coded the data, which prevented the calculation of intercoder reliability. However, the involvement of the facilitator in the development of the coding hierarchy and refinement of the identified themes enhanced the trustworthiness of the resulting interpretation.[30 31] Second, the findings only represent the perspectives of the 139 participants who attended the FGs. Although the size of the sample is large for qualitative research, caution should be exercised when generalising to the broader population. Finally, this research was conducted in one geographic location. Future research could explore perceptions of the effectiveness of e-cigarette regulations in other jurisdictions.

This study had several strengths. To our knowledge, this is the first study to explore perceptions of regulation effectiveness, with prior work in this space assessing regulation

support. Given support for e-cigarette regulations is influenced by vaping status,[9] the present study's exploration of perceived effectiveness offers a more objective account of community views. Second, we were able to obtain unprompted recommendations for effective e-cigarette regulations prior to the government's announcement of the new reforms. This provides a real-world opportunity to assess the link between the perceived effectiveness of reforms and the outcomes of implementation. Finally, we recruited participants across the community and stratified groups by age, gender and vaping status. This allowed us to (1) gather a broader range of perspectives than previous studies and (2) explore differences between groups. Given participants are more likely to feel comfortable sharing their perspectives in a homogeneous group,[32] the stratification also maximised the chances that the discussions held were rich and participant responses were genuine.

## CONCLUSION

Results from the present study exploring perspectives on Australia's e-cigarette regulations and recommendations for further regulations suggest that many believe the current prescription model for nicotine e-cigarette products to be effective when enforced appropriately. Several concerns were raised about the availability of non-nicotine e-cigarettes, suggesting that planned reforms to restrict the supply of these products are likely to be well received. Given many of the recommendations made by participants for effective regulations were those announced by the Federal Government in the months after this study was conducted, the proposed reforms are likely to be met with little objection from the public. However, ensuring the reforms are complemented by efforts to support those who are addicted to e-cigarettes or who are using e-cigarettes to cope with mental health struggles is critical.

**Contributors** MIJ: funding acquisition, conceptualisation, methodology, writing–original manuscript, review and editing, supervision, project administration. SJLY: methodology, writing–reviewing and editing. MEB: data curation, formal analysis, writing–original manuscript, reviewing, editing. MIJ is the guarantor for this work.

**Funding** This work was supported by a National Health and Medical Research Council Investigator Grant (APP1194713).

**Disclaimer** The funding source had no involvement in study design; in the collection, analysis and interpretation of data; in the writing of the article; and in the decision to submit the article for publication.

**Competing interests** None declared.

**Patient and public involvement** Patients and/or the public were not involved in the design, or conduct, or reporting, or dissemination plans of this research.

**Patient consent for publication** Not applicable.

**Ethics approval** This study involves human participants and was approved by the University of Melbourne Human Research Ethics Committee: 2023-24865-38247-10. Participants gave informed consent to participate in the study before taking part.

**Provenance and peer review** Not commissioned; externally peer reviewed.

**Data availability statement** Data are available on reasonable request. Data will not be shared with those affiliated with the tobacco or e-cigarette industries.

**ORCID iDs**
Mary-Ellen E Brierley http://orcid.org/0000-0003-1732-2260
Michelle I Jongenelis http://orcid.org/0000-0002-0717-1692

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
