## [Reviewer comments · BMJ Open]

ARTICLE DETAILS

TITLE (PROVISIONAL)	Perceptions of Australia's e-cigarette regulations and recommendations for future reforms: A qualitative study of adolescents and adults
AUTHORS	Brierley, Mary-Ellen; Yaw, Sean; Jongenelis, Michelle I.

VERSION 1 – REVIEW

REVIEWER	Samia Amin Macquarie University Australian Institute of Health Innovation
REVIEW RETURNED	05-Nov-2023

GENERAL COMMENTS	Thank you for the opportunity to review this timely and important study exploring perceptions of e-cigarette regulation effectiveness. Understanding public views on regulatory measures is critical as policies continue to evolve globally. In the data analysis section, the authors state they used reflexive thematic analysis and refer to the steps outlined by Braun & Clarke (2019). However, they do not provide much detail on how the coding was actually conducted. For example: The initial coding process - Were codes emergent or based on a start list? Please provide examples of early codes. Iterative theme refinement - How exactly did the author (s) review and refine themes? Were any combined or dropped? Coder agreement - Did multiple team members code transcripts and discuss themes? How disagreements were resolved Theme definition - Were theme names descriptive labels or more conceptual? How were they defined? Adding insight into these coding procedures will significantly enhance methodological transparency and rigor. As is, the analysis approach is articulated only very broadly. Providing the additional details above will help readers better evaluate the coding and thematic development process. Overall this is a strong, well-executed study with important implications. Strengthening the write-up of the analysis procedures will further augment the contribution.
--

REVIEWER	15-Dec-2023
REVIEW RETURNED	Bertrand Dautzenberg Hopital Pitie-Salpetriere Service de Pharmacologie

GENERAL COMMENTS	Review Perceptions of Australia's e-cigarette regulations and recommendations for future reforms: A qualitative study of adolescents and adults
--

	General remarks This article addresses an interesting subject on the perception among young people and adults of the regulation of electronic cigarettes in Australia, a country where tobacco consumption is low, and the regulation of nicotine is specific. The introduction lacks information on the evolution of Australian regulations on electronic cigarettes (with and without nicotine) for readers who is not specialist in nicotine use in Australia. □A sentence in the text on the new regulation (with date of real implementation) will be welcome. □A sentence in the text on the broad outlines of consumption since the appearance of the electronic cigarette would also be welcome. To my knowledge, in Australia, electronic cigarettes with nicotine require a prescription for adults. Electronic cigarettes with nicotine are completely prohibited for sale to minors, but it would be useful to have information on the dates of implementation of these measures. Important general remark: your study is built on a unilateral vision of the interaction e-cigarette/cigarette when the interaction in adolescent is multilateral:  1- A gateway effect explains only a very small amount of initiation of smoking (5 à 10%). 2- A hypothetic cessation effect of e-cigarette on cigarette may concern only regular smokers, so don't concern adolescent but only young adult and is very marginal, 3- The main interaction is the diversion effect of e-cigarette on cigarette use but is not considered in your study. Indeed, your introduction states as an established fact that the electronic cigarette is only a gateway to cigarettes for young people without open the hypothesis of competition effect. According to my research, recent consumption figures among young people in Australia show an increase in cigarette use (after 30 years or regression of usage), with the prohibition of electronic cigarettes in adolescent. In your introduction, you do not mention a possible link between the recent ban on the sale of electronic cigarettes to children and the resumption of nicotine consumption among children do not call into question the overall quality of your study although this could bias the results. In detail Abstract The abstract summarizes the content of the article well. You are highlighting your paper that the lack of understanding of Australian regulations regarding e-cigarettes in your summary. It would be good to recall it very briefly in the introduction. Article summary It will be important to specify the date of the study because the responses must be analyzed according to the evolution of Australian regulations. Introduction  • The sentence of the abstract "The need for supply reduction measures to include addiction and mental health supports was discussed" as the introduction suggests that there is a bilateral link between the two whereas if it is well established that mental illness increases the risk of smoking, the opposite is not demonstrated (to my knowledge unless you include that tobacco dependence as a
--	---

	mental health problem). If that's what you mean, provide references.  • Provide the date of prohibition of nicotine for adolescent. • Page 5 second paragraph - You presuppose without argumentation that any measure restricting the availability of electronic cigarettes among adults and adolescents is necessarily 100% favorable to public health. - By doing this you are unaware that the electronic cigarette which is the first means of quitting tobacco (whether it is used as a medicine or as a pleasure product for changing the way an addict takes nicotine), you are unaware that the ability of electronic cigarette to maintain dependence to the nicotine is less than cigarette. You ignore and that most cigarette users switching to electronic cigarettes stop taking any nicotine after 3 months because the need for nicotine has gradually decreased as they would with nicotine substitutes. - You assume that among adolescents the electronic cigarette is only a gateway but is never a competitor to cigarettes while studies now show that the e-cigarette is less of a gateway as a competitor to cigarettes, if you don't analyze only the non-smokers sub-cohorts, but the whole adolescent population.  • You should either clearly state that this is your presupposition and to consider only the subpopulation who initiate e-cigarette first, or you should provide arguments to support your one-sided point of view which does not conform to reality among most adolescents. • As in any behavioral problem, the truth is often bilateral and overly restrictive regulation can promote or reduce cigarette consumption and therefore be favorable or unfavorable to public health, but only one of the two aspects is studied. by your study: It is necessary that you specify your decision either here in the introduction, or in the discussion. Procedure  • Precise what are "Government's announcement" of March 2023 • Who is XX ?(Principal research fellow ? One of the authors ?). Data analysis  • Who is XX ? • You don't provide any information on cigarette consumption evolution (Is This figure of increase of tobacco cigarette consumption in adolescent a problem? What is hypothesis on the link between the discussion then the implementation of the AU new law and this historical increase of tobacco consumption? (To be present in introduction or discussion but important for the interpretation of yours results Results No comments Discussion  • A explanation on prescription model is need:
--	---

	- Prescription only by Medical doctor or also nurses midwives... - Prescription only for cessation or also for a long-term nicotine replacement - Who pay the nicotine (User or insurance?)  • Some data on e-cigarette with nicotine, e-cigarette with non-nicotine and tobacco cigarette use will be welcome. • You said many in the sample questioned why these products existed and their utility, but there is no question on tobacco cigarette, The possible utility of these products exists if cigarette is available (The utility is different in presence and in absence of cigarette on the market). Limitations and strengths  • Page 15 one line before the end you speak of regulation effectiveness, but you have to precise what mean effectiveness.  o Effectiveness to stop e-cigarette or effectiveness to end tobacco (The E-cigarette is one of product used by smoker to qui tobacco!) Conclusions  • Page 16 line 39 (most of long-term users of e-cigarette are tobacco smokers, and the dependance is mainly a persisting nicotine dependance induced by cigarette, not an addiction created by e-cigarettes.
--	---

VERSION 1 – AUTHOR RESPONSE

Reviewer feedback	Authors' responses
Reviewer 1	
1. Thank you for the opportunity to review this timely and important study exploring perceptions of e-cigarette regulation effectiveness. Understanding public views on regulatory measures is critical as policies continue to evolve globally.	Thank you for this positive feedback.
2. In the data analysis section, the authors state they used reflexive thematic analysis and refer to the steps outlined by Braun & Clarke (2019). However, they do not provide much detail on how the coding was actually conducted. For example: The initial coding process - Were codes emergent or based on a start list? Please provide examples of early codes. Iterative theme refinement - How exactly did the author (s) review and refine themes? Were any combined or dropped?	We have now provided more detailed information on the coding and theme construction process in the data analysis section of the manuscript (Paragraphs 1, 2 and 3). We have also included the final coding hierarchy in the online supplementary material. Pg 7: As this research was data-driven rather than theory-driven, an inductive (i.e., emergent) approach to thematic analysis was adopted (Braun & Clarke, 2006). Pg 8: Initial codes were generated that organised the data into the following broad topics: (i) beliefs about current regulations and (ii) recommendations for potential regulations. For the former, child

	codes were created that further organised the data into the topics of (i) beliefs about the prescription model and (ii) beliefs about the availability of non-nicotine e-cigarette products. Within each of these child codes, further codes were generated that organised the data based on (i) positive sentiment and (ii) negative sentiment. In terms of regulation recommendations, child codes were created for each of the ideas generated by participants (e.g., price increases, plain packaging, product warning labels). Codes featuring similar content were then merged (e.g., 'plain packaging' and 'product warning labels' were merged to form a 'product packaging' code). These codes were then refined into the categories of (i) supply factors and (ii) demand factors. Descriptive labels were assigned to all codes. Please see supplementary material for the final coding hierarchy.
3. Coder agreement - Did multiple team members code transcripts and discuss themes? How disagreements were resolved	The data analysis section of the original manuscript noted that coding was conducted by a single researcher (as is customary in reflexive thematic analysis). We have made amendments to this section to clarify that regular meetings were held between members of the research team: Pg 7: This included data familiarisation, code generation, theme generation, theme review, defining and naming themes, and combining the analysis of themes in the context of a report (Braun & Clarke, 2019). Throughout this process, a series of 'critical friends' meetings was held (Smith & McGannon, 2018). During these meetings, XX and YY reviewed the coding hierarchy, refined the identified codes, and generated key

	themes based on these codes. The limitations section of the manuscript also includes the following: Pg 16: First, due to the emergent nature of the coding process, only one researcher coded the data, which prevented the calculation of inter-coder reliability. However, the involvement of the facilitator in the development of the coding hierarchy and refinement of the identified themes enhanced the trustworthiness of the resulting interpretation (Elo et al., 2014; Nowell, Norris, White, & Moules, 2017).
4. Theme definition - Were theme names descriptive labels or more conceptual? How were they defined?	Please see Point 2.
5. Adding insight into these coding procedures will significantly enhance methodological transparency and rigor. As is, the analysis approach is articulated only very broadly. Providing the additional details above will help readers better evaluate the coding and thematic development process. Overall this is a strong, well-executed study with important implications. Strengthening the write-up of the analysis procedures will further augment the contribution.	Thank you for your constructive comments, which we believe have allowed us to significantly improve the manuscript.
Reviewer 2	
6. General remarks: This article addresses an interesting subject on the perception among young people and adults of the regulation of electronic cigarettes in Australia, a country where tobacco consumption is low, and the regulation of nicotine is specific.	Thank you for this positive feedback.
7. The introduction lacks information on the evolution of Australian regulations on electronic cigarettes (with and without nicotine) for readers who is not specialist in nicotine use in Australia. A sentence in the text on the new regulation (with date of real implementation) will be welcome. A sentence in the text on the broad outlines of consumption since the appearance of the electronic cigarette would also be welcome. To my knowledge, in Australia, electronic cigarettes with	Thank you for this feedback. The Australian Government has only recently (28th November 2023) announced the dates of implementation for the new regulations. These have now been provided: Pg 5: To address these issues, Australia's Federal Government announced plans in May 2023 to introduce regulations that prohibit the importation of both nicotine and

nicotine require a prescription for adults. Electronic cigarettes with nicotine are completely prohibited for sale to minors, but it would be useful to have information on the dates of implementation of these measures.

non-nicotine e-cigarettes for non-therapeutic use (Department of Health and Aged Care, 2023a). Restrictions on flavourings and the introduction of pharmaceutical-like packaging were also proposed to reduce the appeal of e-cigarette products. *These regulations will be implemented on the 1st March 2024 (Therapeutic Goods Administration, 2023).*

As suggested by this reviewer, a statement that broadly outlines e-cigarette consumption in Australia since the appearance of the devices on the market has now been provided:

Pg 4: In Australia, the context of the present study, *e-cigarette use has increased significantly since reporting measures were introduced in 2013 (Greenhalgh, Bain, Jenkins, & Scollo, 2023), with the number of adolescents and young adults that report having used an e-cigarette in the past month increasing approximately five-fold since 2018 (Wakefield, Haynes, Tabbakh, Scollo, & Durkin, 2023).*

A summary of Australia's current regulations was provided in our original submission (Page 4). We have expanded this content and now provide further information detailing the date the current regulations were implemented. We have also made it clear that e-cigarettes both with and without nicotine have always been illegal to supply to those < 18 years:

Pg 4: *Since the 1st October 2021, nicotine-containing e-cigarettes and related products have been legally available to adults only via*

	prescription from a medical doctor for the purposes of smoking cessation (Greenhalgh et al., 2022). E-cigarettes that do not contain nicotine are less restricted and may be sold by retailers to those aged 18+ years in all states and territories except Western Australia (Greenhalgh et al., 2022). The supply of e-cigarettes – regardless of nicotine content – to individuals under 18 years of age has never been permitted.
8. Important general remark: your study is built on a unilateral vision of the interaction e-cigarette/cigarette when the interaction in adolescent is multilateral:  1- A gateway effect explains only a very small amount of initiation of smoking (5 à 10%). 2- A hypothetic cessation effect of e-cigarette on cigarette may concern only regular smokers, so don't concern adolescent but only young adult and is very marginal, 3- The main interaction is the diversion effect of e-cigarette on cigarette use but is not considered in your study. Indeed, your introduction states as an established fact that the electronic cigarette is only a gateway to cigarettes for young people without open the hypothesis of competition effect. According to my research, recent consumption figures among young people in Australia show an increase in cigarette use (after 30 years or regression of usage), with the prohibition of electronic cigarettes in adolescent. In your introduction, you do not mention a possible link between the recent ban on the sale of electronic cigarettes to children and the resumption of nicotine consumption among children do not call into question the overall quality of your study although this could bias the results.	Thank you for these general remarks. We have not referred to the gateway hypothesis in the manuscript and have not based our rationale for the current study on the gateway hypothesis. It is also incorrect to state that we report the gateway effect as an “established fact”. Rather, we have stated e-cigarette use has been found to be associated with tobacco smoking initiation. This statement is based on the results from the most comprehensive systematic review and meta-analysis of the worldwide evidence on the relationship between e-cigarette use and uptake of smoking conducted to date (Banks et al., 2023). This review concluded that (pp. 364-365): “There is substantial and consistent evidence from observational studies that never smokers who have used e-cigarettes are more likely than those who have not used e-cigarettes to try smoking conventional cigarettes and to transition to becoming regular tobacco smokers.” “Based on strong evidence, never smokers who use e-cigarettes are on average around three times as likely as those who do not use e-cigarettes to initiate cigarette smoking.” “There is strong evidence that non-

	smokers who use e-cigarettes are also around three times as likely as those who do not use e-cigarettes to become current cigarette smokers.” We have been unable to locate any evidence of a ‘diversion effect’. The research to which this reviewer refers (Dautzenberg et al., 2023) cannot make conclusions about the diversion effect, as noted by the authors of this manuscript: “Excluding initial tobacco smokers, the sub-cohorts represent only a small fraction of the whole population: they explain a maximum of 5.3% of tobacco smoking initiation in adolescents, assessing only one way of the interaction (the Gateway effect), and exclude any possibility of identifying and quantifying a Diversion effect of cigarettes by e-cigarettes.”(Dautzenberg et al., 2023, pg. 16).” We are unsure of the reviewer’s statement relating to the “recent ban on the sale of electronic cigarettes to children” as the use of e-cigarettes among children has always been prohibited in Australia. As such, there is no “recent ban”. Reference: Banks, E., Yazidjoglou, A., Brown, S., Nguyen, M., Martin, M., Beckwith, K., . . . Joshy, G. (2023). Electronic cigarettes and health outcomes: umbrella and systematic review of the global evidence. Medical Journal of Australia, 218(6), 267-275. doi:https://doi.org/10.5694/mja2.51890
9. Abstract: The abstract summarizes the content of the article well.	Thank you for this positive feedback.
10. You are highlighting your paper that the lack of understanding of Australian regulations regarding e-cigarettes in your summary. It would be good to recall it	Please see Point 7.

very briefly in the introduction.	
11. Article summary: It will be important to specify the date of the study because the responses must be analyzed according to the evolution of Australian regulations.	Details of the study date were provided in the original manuscript. This information remains in our resubmission: Pg 7: Focus groups were conducted in March 2023 (prior to the Government’s announcement) and were approximately 70 minutes in duration (range: 57 to 88 minutes).
12. Introduction: The sentence of the abstract “The need for supply reduction measures to include addiction and mental health supports was discussed” as the introduction suggests that there is a bilateral link between the two whereas if it is well established that mental illness increases the risk of smoking, the opposite is not demonstrated (to my knowledge unless you include that tobacco dependence as a mental health problem). If that’s what you mean, provide references.	Please note that this sentence is placed within the results section of the abstract and refers to the discussions of the focus group participants. It is thus inappropriate to provide references here.
13. Provide the date of prohibition of nicotine for adolescent.	As noted in Point 7, we have added a sentence to Paragraph 2 of the Introduction specifying laws for adolescents. Pg 4: E-cigarettes that do not contain nicotine are less restricted and may be sold by retailers to those aged 18+ years in all states and territories except Western Australia (Greenhalgh et al., 2022). The supply of e-cigarettes – regardless of nicotine content – to individuals under 18 years of age has never been permitted.
14. Page 5 second paragraph - You presuppose without argumentation that any measure restricting the availability of electronic cigarettes among adults and adolescents is necessarily 100% favorable to public health. By doing this you are unaware that the electronic cigarette which is the first means of quitting tobacco (whether it is used as a medicine or as a pleasure product for changing the way an addict	Apologies, but we cannot locate any text where we have noted that restricting the availability of electronic cigarettes among adults and adolescents is “necessarily 100% favorable to public health”. We are aware of the evidence that shows e-cigarettes may be effective smoking cessation aids. However, this

takes nicotine), you are unaware that the ability of electronic cigarette to maintains dependence to the nicotine is less than cigarette. You ignore and that most cigarette users switching to electronic cigarettes stop taking any nicotine after 3 months because the need for nicotine has gradually decreased as they would with nicotine substitutes.  - You assume that among adolescents the electronic cigarette is only a gateway but is never a competitor to cigarettes while studies now show that the e-cigarette is less of a gateway as a competitor to cigarettes, if you don't analyze only the non-smokers sub-cohorts, but the whole adolescent population. - You should either clearly state that this is your presupposition and to consider only the subpopulation who initiate e-cigarette first, or you should provide arguments to support your one-sided point of view which does not conform to reality among most adolescents. - As in any behavioral problem, the truth is often bilateral and overly restrictive regulation can promote or reduce cigarette consumption and therefore be favorable or unfavorable to public health, but only one of the two aspects is studied. by your study: It is necessary that you specify your decision it either here in the introduction, or in the discussion. 	is unrelated to the present study, which aimed to examine the views of Australians towards regulations relating to e-cigarettes. In addition, please note that e-cigarettes are not “the first means of quitting tobacco”. In Australia, e-cigarette use is a second-line treatment for tobacco dependence, with approved nicotine replacement therapies and pharmacotherapies plus behavioural support considered first line treatment (Royal Australian College of General Practitioners, 2021). Please see Point 8 for our response related to the gateway effect. Finally, please note that our study did not aim to explore the merits of restrictive vs liberal e-cigarette regulation, nor do we state that restrictive regulations are the preferred option. Rather, the study explores Australians’ perceptions of current e-cigarette regulations and their suggestions for further regulatory actions that may assist in reducing e-cigarette use among adolescents and those who have never smoked. Accordingly, we are unaware of the “decision” this reviewer states we have made. We welcome editorial guidance on the points raised by this reviewer. We note that Reviewer 1 raised no such concerns. Reference: Royal Australian College of General Practice. (2021). Pharmacotherapy for smoking cessation. Available from: https://www.racgp.org.au/clinical-resources/clinical-guidelines/key-racgp-guidelines/view-all-racgp-guidelines/supporting-smoking-cessation/pharmacotherapy-for-smoking-cessation
15. Procedure: Precise what are “Government’s announcement” of March 2023	There seems to be some confusion. As noted in the manuscript, the Government’s announcement was

	made in May 2023 . The details of this announcement were provided in the original manuscript and remain in the resubmitted manuscript (Pages 4-5).
16. Who is XX ?(Principal research fellow? One of the authors?).	Author initials are blinded for review as is consistent with standard peer-review practice. We have changed the initials for the coder to YY to distinguish between two of the authors referred to in this manuscript.
17. Data analysis: Who is XX ?	Information on the evolution of cigarette consumption in Australia, has now been added to the manuscript:
18. You don't provide any information on cigarette consumption evolution (Is This figure of increase of tobacco cigarette consumption in adolescent a problem? What is hypothesis on the link between the discussion then the implementation of the AU new low and this historical increase of tobacco consumption? (To be present in introduction or discussion but important for the interpretation of yours results Figure 4: Six-monthly prevalence of current smoking by age group, 2018 to 2023 (weighted %).	Information on the evolution of cigarette consumption in Australia, has now been added to the manuscript: Pg 4: The increasing prevalence of e-cigarette use has largely occurred in the context of decreasing tobacco cigarette use, although recent data has observed an increase in tobacco smoking among adolescents and young adults for the first time in three decades (Wakefield et al., 2023).
19. Results: No comments	Thank you.
20. Discussion: A explanation on prescription model is need:  - Prescription only by Medical doctor or also nurses midwives... - Prescription only for cessation or also for a long-term nicotine replacement - Who pay the nicotine (User or insurance?) 	We agree that an explanation of the prescription model is important and provided this in the original submission (Introduction, Paragraph 2). We have made minor revisions to this section to stipulate that medical doctors can prescribe e-cigarettes for smoking cessation purposes: Pg 4: Since the 1st October 2021, nicotine-containing e-cigarettes and related products have been legally available to adults only via prescription from a medical doctor

	for the purposes of smoking cessation (Greenhalgh et al., 2022). A detailed discussion of Australia's healthcare system is beyond the scope of this manuscript.
21. Some data on e-cigarette with nicotine, e-cigarette with non-nicotine and tobacco cigarette use will be welcome.	We provided data on the prevalence of e-cigarette use for the population group of interest in the original manuscript (Introduction, Paragraph 1). To accommodate this reviewer's request regarding tobacco cigarette use, a sentence has been added to the end of this paragraph: Pg 4: Recent years have seen global increases in the prevalence of e-cigarette use, particularly among youth (Australian Institute of Health and Welfare, 2020; Filippidis, Lavery, Gerovasili, & Vardavas, 2017; Obisesan et al., 2020). In Australia, the context of the present study, e-cigarette use has increased significantly since reporting measures were introduced in 2013 (Greenhalgh, Bain, Jenkins, & Scollo, 2023), with the number of adolescents and young adults that report having used an e-cigarette in the past month increasing approximately five-fold since 2018 (Wakefield, Haynes, Tabbakh, Scollo, & Durkin, 2023). These increases are concerning given vaping has been found to be associated with several physical and mental health harms and subsequent initiation of tobacco cigarette smoking (Banks et al., 2023; Berry et al., 2019; Lechner, Janssen, Kahler, Audrain-McGovern, & Leventhal, 2017). The increasing prevalence of e-cigarette use has largely occurred in the context of decreasing tobacco cigarette use, although recent data has observed an increase in tobacco smoking among adolescents and young adults for the first time in three decades

	(Wakefield et al., 2023). Please note that national surveys in Australia do not distinguish between nicotine and non-nicotine e-cigarette use and we are therefore unable to provide this information.
22. You said many in the sample questioned why these products existed and their utility, but there is no question on tobacco cigarette, The possible utility of these products exists if cigarette is available (The utility is different in presence and in absence of cigarette on the market).	There appears to be some confusion. Participants did not question the reasons for e-cigarette availability, but the utility of non-nicotine products. Please see relevant text from the manuscript below (Discussion, Paragraph 4): Pg 15: The concerns raised regarding non-nicotine e-cigarettes, and the finding that many in the sample questioned why these products existed and their utility, suggests that plans to restrict access to such products will likely be accepted by the general public.
23. Limitations and strengths: Page 15 one line before the end you speak of regulation effectiveness, but you have to precise what mean effectiveness. Effectiveness to stop e-cigarette or effectiveness to end tobacco (The E-cigarette is one of product used by smoker to qui tobacco!)	There appears to be some confusion. As per the aim of the study, the discussion section presents content on participants' perceptions of regulation effectiveness, not actual regulation effectiveness. We have provided a description of perceived effectiveness in the Introduction of this manuscript (please see Paragraph 5). As stated throughout the manuscript, we assessed perceptions of the effectiveness of regulations at minimising e-cigarette use, particularly among adolescents and those who have never smoked.
24. Conclusions: Page 16 line 39 (most of long-term users of e-cigarette are tobacco smokers, and the dependance is mainly a persisting nicotine dependance induced by cigarette, not an addiction created by e-cigarettes.	Thank you for this information.

VERSION 2 – REVIEW

REVIEWER	Samia Amin Macquarie University Australian Institute of Health Innovation
REVIEW RETURNED	21-Jan-2024

GENERAL COMMENTS	I appreciate the opportunity to review the revised version. It's noteworthy that the author has incorporated a significant methodological update in response to suggestions from the earlier review process. I have no further comments. This research has the potential to advance our comprehension of e-cigarette regulation effectiveness in Australia, potentially influencing current perceptions on the subject.
--

REVIEWER	Bertrand Dautzenberg Hopital Pitie-Salpetriere Service de Pharmacologie
REVIEW RETURNED	19-Jan-2024

GENERAL COMMENTS	Review is OK
--------------